# The Impact of Eucalyptus and Pine Plantations on the Taxonomic and Functional Diversity of Dung Beetles (Coleoptera: Scarabaeidae) in the Southern Region of Ecuador

**DOI:** 10.3390/biology13100841

**Published:** 2024-10-19

**Authors:** Karen Sanmartín-Vivar, Jessica Guachizaca-Macas, Diego Marín-Armijos

**Affiliations:** Colección de Invertebrados Sur del Ecuador, Museo de Zoología CISEC-MUTPL, Departamento de Ciencias Biológicas y Agropecuarias, Universidad Técnica Particular de Loja, Loja 110160, Ecuador; kjsanmartin@utpl.edu.ec (K.S.-V.); jmguachizaca@utpl.edu.ec (J.G.-M.)

**Keywords:** biodiversity, land use, habitat degradation, ecosystem services, forest plantations, conservation

## Abstract

**Simple Summary:**

This study analyzed the effect of forest plantations on the taxonomic and functional diversity of dung beetles in comparison to natural forests in southern Ecuador. The results indicated a greater abundance of generalist species in the plantations, likely due to their high level of adaptation to disturbances and their presence in nearby forests. This could be attributed to the proximity of these areas to urban zones, which may influence the dung beetle community, as their primary food source depends on mammals. However, this study highlights the lack of information on this topic in the Neotropics, particularly in Ecuador, suggesting that further research is needed to compare and validate these findings.

**Abstract:**

This study sheds light on the complex relationship between land use, biodiversity, and the functional traits of dung beetles in Ecuador. The results indicate that the richness and abundance of dung beetles vary across different land uses and regions, with forests generally having a positive impact, while eucalyptus and pine plantations have a negative effect in certain areas. Specific indicator species, such as *Homocopris buckleyi* for forest areas and *Onthophagus curvicornis* for eucalyptus plantations, were identified. This study also found that functional diversity analysis, based on morphological traits, revealed that certain traits, such as biomass, pronotum width, head width, and elytra length, were significant contributors to differences in dung beetle communities across various land uses and regions. This study highlights the potential conservation value of certain modified habitats and emphasizes the importance of considering both taxonomic and functional diversity when assessing the impact of land use on the ecosystem services provided by dung beetles. It underscores the potential value of plantations as refuges for dung beetle communities and the need for long-term assessments to better understand biodiversity changes over time.

## 1. Introduction

Biodiversity is the property in which living systems are different and provide certain environmental services through which humans obtain various benefits [1,2,3]. However, there is currently a progressive degradation and depletion of biological systems and their diversity, owing to human activities and strategies [4]. In addition, it is threatened by deforestation and fragmentation, as well as other causes, such as the introduction of exotic species, land use change, and climate change, which can compromise important ecosystem functions and services, as well as the structure of the landscape [5,6]. This drives a range of environmental changes that can affect the suitability of habitats for wildlife [7,8,9].

Human impact on Earth has caused a considerable global decline in species richness worldwide, such as dung beetles, which a study found to have an overall negative effect on their richness and abundance [10]. However, Salomão et al. [11] suggest that land use does not affect all organisms in terrestrial ecological communities equally and that various functional groups of species may respond differently.

Dung beetles are key elements in ecosystems because they play diverse ecological roles in nutrient recycling, soil aeration, seed dispersal, fly control and dung removal [12,13,14]. Some studies have evaluated that land use change affects the ecological functions they perform in neotropical ecosystems and similar results were found in Raine et al.’s study [15]. Additionally, according to [16], the relocation of dung by beetles brings about a series of processes that significantly impact the functionality of the ecosystem, which is considered an excellent indicator of this process. Several works have proposed the study of beetle diversity patterns in forest plantations, in which it has been highlighted that each type of plantation has multiple impacts on diversity patterns and also significantly influences their richness, abundance and diversity [17,18]. As mentioned by López-Bedoya et al. [19], the recent global meta-analysis shows that forest plantations have a negative effect on this group, so it is important to know their current situation. However, there are currently few studies on the effects of different land uses on the functional traits of beetles [20]. Therefore, it is very important to analyze their situation through traits [21] since they respond to environmental stress and changes in the ecosystem [22,23].

For the tropical region, there is insufficient research on dung beetles; therefore, [19] recommends research focused on the conversion of forests to other land uses to learn how this group responds to habitat change, especially given the high levels of deforestation that have reduced forests by 75% in the tropics [24].

Therefore, the present study aimed to analyze the effect of land use on taxonomic and functional diversity based on morphological traits of the dung beetle community. We set the following objective: “to evaluate the effect of land use on the taxonomic and functional diversity of dung beetles”. We expected a higher abundance and diversity of dung beetles in the forest since the results of other studies carried out in Brazil indicate that land use change taxonomically affects dung beetles due to the lower number of species found in eucalyptus plantations compared to natural forest [25].

## 2. Materials and Methods

### 2.1. Study Area

The research was carried out in two of the areas with the greatest distribution of pine and eucalyptus plantations in southern Ecuador. Both eucalyptus (*Eucalyptus globulus* Labill.) and pine (*Pinus patula* Schiede ex Schltdl. & Cham.) plantations are known worldwide as introduced species [26]. These cover approximately 164,000 ha, 90% of which are distributed in the inter-Andean region, 8% in the coast and 2% in the Amazon [27]. These species were used for this study because they are the two main species that have been reforested since the 1970s. [27,28]

The province of Loja is influenced by its geographical location and the Huancabamba Depression, which facilitates the development of fauna, flora and ecosystems [29]. It has a variety of climates: high mountain equatorial, dry mesothermal equatorial and semi-humid mesothermal equatorial. Rainfall varies between 758 and 1250 mm. The rainy season lasts from November to May. Relative humidity fluctuates between 80% and 88%, and the temperature ranges between 8 and 27 degrees Celsius.

We selected two locations at least 1 km apart, and three land uses were selected: (1) forest, (2) eucalyptus plantation and (3) pine plantation (Figure 1; Table 1).

### 2.2. Dung Beetle Sample

In each land use, 27 pitfall traps were placed in three transects of 300 m each, and these were arranged in a triangular shape separated by a distance of 1 m at 0, 150 and 300 m and remained in the field for 48 h. The traps consisted of plastic containers 12 cm in diameter and 9 cm deep and were placed at ground level with a mixture of water and detergent. Each trap was baited with 30 g of decomposing pig manure. The captured samples were then placed in polyethylene bags containing 90% ethyl alcohol. For this sample, we tried to follow the methodologies proposed by Nunes et al. and Mora-Aguilar et al. [9,30] for the Neotropics. In addition, these types of traps were used because they are considered to be the dominant method for capturing dung beetles [30].

The samples were then separated and identified to the species level in the laboratory of the Museum of Zoology (CISEC-MUTPL) using taxonomic keys according to Chamorro et al. and Chamorro et al. [31,32] and compared with the reference collection of the “Colección de Invertebrados Sur del Ecuador CISEC-MUTPL”.

### 2.3. Morphometric Measurement

To characterize functional diversity based on traits, we randomly selected and measured five individuals per species and per land use [33,34]. We selected nine morphological traits based on bibliographical research [4,33,34,35]. Measured morphological features: 1. HL = head length, 2. HW = head width, 3. PL = pronotum length, 4. PW = pronotum width, 5. PH = pronotum height, 6. EL = length of elytra, 7. pTL = length of the protibia, 8. pTW = width of the protibia, and 9. mTL = length of metatibia (Figure 2).

Biomass was measured by species and for the entire assemblage in each land use, using the following formula [36]:$$biomass\;(in\;g)=0.305\times \frac{L^{2.62}}{1000}$$
where *L* (length) is the sum of the head length, pronotum length, and elytra length of the individual and/or species sampled in millimeters.

#### 2.3.1. Taxonomic Diversity

The dung beetle community was analyzed using hill numbers for each land use for data based on richness (q = 0) and diversity (q = 1 and q = 2) [37]. We also performed non-metric multidimensional analysis (NMDS), based on the Bray–Curtis index, using the abundance data obtained to graphically express the grouping patterns and differences in species composition.

The INDVAL species indicator was used to select habitat indicator species, which is based on the degree of specificity (exclusive to a particular habitat) and fidelity (the frequency of occurrences within the same habitat) of the species [38].

To evaluate the abundance distribution, we performed a rank abundance curve, which showed the relative abundance of species and their uniformity within each land use, providing essential information for predicting changes in community patterns and allowing us to better explain how the dominance and distribution of species behaved in each land use [18].

To evaluate the effects of land use (forest, pine, and eucalyptus plantations) on the composition of dung beetles, Generalized Linear Models (GLMs) were applied using the Poisson distribution in the R 4.2.2 software package [39], which helped measure the incidence of the three land uses to the number of coprophagous beetles found in the sampling.

#### 2.3.2. Functional Diversity Based on Morphological Traits

The analysis of functional traits is important since it is linked to the functions that species fulfill in ecosystems, such as locomotion, dispersal capacity, etc. [21]. Therefore, for the simple reduction in the data sets with graphical representation and to position the species in the functional trait space, principal component analysis (PCA) was performed using R 4.2.2 software [39].

A linear mixed model was used to evaluate the effects of land use (forest, pine, and eucalyptus plantations) on the morphological traits of dung beetles. It allows multiple nested or crossed random effects, computes profile confidence intervals, and conducts parametric bootstrapping [40]. Sites were included as random intercepts to avoid potential non-independence among land use types within the same area. All analyses were performed using the R 4.2.2 software [39].

## 3. Results

### 3.1. Taxonomical Diversity

In total, we sampled 1616 dung beetles, corresponding to 10 species and 8 genera (Appendix A). Seven species were collected from forests and pines, and six were collected from eucalyptus. The most abundant species in the forest were *Uroxys* sp. 2 (28.0 ± 34.8) and *Homocopris buckleyi* (6.94 ± 4.01). In pine plantations, the most abundant species were *Uroxys* sp. 1 (120.5 ± 129.0) and *Uroxys* sp. 2 (25.5 ± 24.0). In eucalyptus plantations, the most abundant species were *Uroxys* sp. 2 (24.8 ± 51.3), followed by *Onthophagus curvicornis* (15.0 ± 4.5). Dung beetles were more abundant in pine plantations (n = 975) and less abundant in eucalyptus plantations (n = 267). Between areas, area 1 (Loja) showed the highest richness (n = 8) and abundance (n = 1455), representing 80% of coprophagous fauna and 90% of individuals (Table 2). In area 1 (Loja), *Uroxys* sp. 2 constituted 65% of the total abundance and was exclusively found in this location, as well as *Uroxys* sp. 1 with the 11% and *Onthophagus curvicornis* with 12% of total abundance present in both areas. Singleton species represented 0.2%, and doubletons represented 2.5%.

Sampling coverage was suitable for most land uses, except pinus and eucalyptus plantations in area 1 (99%) and pinus plantations (95%) in area 2 (Table 3). According to the rarefaction and extrapolation curves, species richness (q = 0) was higher in plantations in both areas, and forest richness was the lowest. The Shannon diversity (q = 1) was higher in the forest in area 1 and in area 2 plantations. The Simpson diversity (q = 2) was higher in the forest in area 1, and in area 2, plantations were higher (Table 3).

With an analysis of non-parametric multidimensional scaling (NMDS), the coprophagous beetle community in area 1 (Loja) was different for forests, but the plantations presented similar communities (r2 = 0.16; *p* < 0.001) (Figure 3a). In area 2 (Saraguro), the community areas were similar and overlapped (r2 = 0. 45; *p* < 0.002) (Figure 3b).

According to the IndVal species indicator, *Homocopris buckleyi* (IndVal = 0.439; *p* < 0.011) was the forest indicator species, in contrast to *Onthophagus curvicornis* (IndVal = 0.561; *p* < 0.010). However, no indicator species were found in pine.

The abundance distribution was more uniform in the forest within area 1 (Loja). Other land uses (eucalyptus and pinus) were the dominant species in both areas. *O. curvicornis* could be considered a common dominant species presented in plantations in both areas. *Uroxys* sp. 2 was dominant in area 1 for all land use types. *H. buckleyi* could be considered a specialist species found only in the forest within area 1 (Figure 4).

In area 1, land use forest (Z = 55.856, *p* < 0.001) and pinus (Z = 19.323, *p* < 0.001) had a positive effect on dung beetle abundance (Table 4). In area 2, land use forest (Z = 33.950, *p* < 0.001) had a positive effect on dung beetle abundance, but in eucalyptus (Z = −6.149, *p* < 0.001) and pinus (Z = −6.910, *p* < 0.001), the effect was negative on dung beetle abundance (Table 4).

### 3.2. Functional Diversity

In the principal components analysis (PCA), biomass is a highly contributing and significant trait in terms of the difference between land uses, as well as pronotum width (PW) and elytra length (EL), unlike protibil width (pTW), which is the least significant trait (Figure 5).

The results of the PCA of the 10 traits analyzed for both areas (Appendix A) showed that all the morphological traits were responsible for the largest degree of variation in PC1 (area 1 = 85.5%, area 2 = 91.2%), whereas for PCA2, only biomass (area 1 = 9.2%, area 2 = 8.6%) was responsible (Figure 5a,b). According to the contribution of area 1, head width (HW) and biomass had the greatest contribution (Figure 5a), and for area 2, only biomass did. In agreement with morphological traits, we found a high correlation of most morphological traits in both areas (Figure 5a,b), mainly in area 2.

We did not find significant effects of land use on measures of morphological traits in either area (Table 5). However, in areas 1 and 2, most traits showed significant values, except for the protibial width (pTW) in area 2 (Table 5). With respect to species in area 1, all of them showed significant values, except for *Uroxys* sp. 2. In area 2, only *Onthophagus curvicornis* presented a significant value (Table 5).

## 4. Discussion

Our results describe the effect of land use on taxonomic and functional diversity through traits and indicate that its richness was similar and poor for each type of use; the results are homogeneous in most land uses by area, which responds to anthropogenic activities [41]. This could be dangerous from a taxonomic point of view as there are many species that remain undescribed, and their distribution and abundance remain unknown.

It is established that both forests and pine plantations harbor the same species richness, but a higher abundance is shown in pine plantations, despite the fact that some studies have shown that dung beetles respond negatively to habitat alterations [42]. Some species, such as *Homocopris buckleyi*, are limited to occupying forests, while others, such as *Onthophagus curvicornis*, can inhabit modified habitats. Previous studies have shown that *O. curvicornis* inhabits these ecosystems as well as *Uroxys* sp. [43], as plantations give rise to an open niche for generalist species.

In our study, it was evident that the results differed from these findings, as did those of Braga et al. and Gardner et al. [44,45], where they found that intense use and soil modification caused a change in the richness and abundance of beetle species [46]. This is possibly due to the wide canopy of the pine plantation, which helps to reduce the temperature of the soil. On the other hand, the decline of beetles in eucalyptus plantations can be explained by the relatively open canopies in warm, dry environments [45]. This is related to the water-holding capacity of eucalyptus roots, which decreases soil moisture and hinders the lifecycle of beetles [28].

Several factors may account for the similarity of dung beetle communities in plantations across both regions, in contrast to the unique communities observed in forests. The homogeneous nature of eucalyptus and pine monocultures in plantations often results in reduced habitat diversity [47], creating comparable ecological conditions across different plantation sites. This uniformity may support similar dung beetle species in both areas. Plantations typically lack the complex structure, microhabitats, and diverse plant associations characteristic of forests, leading to a convergence of species adapted to simplified environments [48,49]. Forests, on the other hand, offer a broader range of niches due to their greater structural diversity, including variations in plant species composition, canopy coverage, understory complexity, and the availability of resources like decomposing organic matter [50]. These factors likely create distinct microenvironments that support more specialized dung beetle communities in each forest area. As a result, forest dung beetle assemblages remain more distinct due to the variability in habitat conditions and resource availability, while the more uniform conditions in plantations lead to community overlap.

Species with specific requirements, such as *Homocopris buckleyi*, rely heavily on particular environmental conditions or resources. The occurrence of these organisms in wooded regions suggests that these environments offer unique ecological niches supporting species with limited habitat flexibility [43]. As a result, protecting these specialists is vital, given that their existence hinges on the conservation of specific ecosystems. Research indicates that forests, especially those at higher elevations, contain distinct dung beetle communities, including specialist species [51,52]. These discoveries underscore the importance of safeguarding forest habitats, particularly those that sustain specialist organisms. Moreover, it is crucial to maintain connections between forest fragments to facilitate movement and genetic exchange among specialist species populations, preventing isolation and localized extinctions [53]. Conservation strategies should be implemented to preserve habitat diversity, ensuring the protection of microhabitats and preventing human activities like logging or land conversion from reducing ecological complexity [54,55,56]. Promoting the use of agroforestry practices or more diverse plantation management techniques that incorporate native species can enhance biodiversity and lessen the negative effects of monoculture plantations on specialist species [57,58]. Establishing ongoing biodiversity monitoring programs that focus on dung beetle communities, especially specialist species, is essential to ensure the effectiveness and sustainability of conservation efforts [30,59].

Functional diversity based on morphological traits is a complementary analysis of taxonomic diversity [35,54]. Therefore, the variables with the highest contribution are PW (pronotum width), pTW (protibia width), EL (elytra length) and biomass. The first three (PW, pTW, and EL) are associated with excavation and dispersal capacity [9], and with respect to biomass, it is a highly contributory trait mainly in pine plantations, although food is a limited resource in plantations [60]. Our study differs from [61], in which they showed that other types of land cover have a negative effect by decreasing the biomass of dung beetles in relation to forests; this is possibly related to the availability of resources since the site is located near an urban area, which probably affects the presence of urban fauna.

In this study, dung beetles were shown to be composed of different species with different habitat ranges. Some cannot pass through altered habitats, while others occupy specific ecosystems [62]. Based on our study, the results differ in the two areas and this is possibly associated with altitude since, when it increases, richness and abundance decrease [43]. In addition, it can be explained that similar levels of diversity could be due to the fact that the referenced forests present disturbances because they are close to an urban area, as in the case of area 1, which does not allow the establishment of a greater number of species.

These findings indicate the need for more research on the conversion of forests for other land uses such as forest plantations, in addition to suggesting other studies that evaluate high mountain ecosystems, since most of the research in the Neotropics is carried out at low elevations, as suggested by López-Bedoya et al. [19]. In Ecuador, significant information gaps persist regarding dung beetles.

## 5. Conclusions

In this study, the effects of forest plantations on the taxonomic and functional diversity of dung beetles in the southern region of Ecuador were evaluated, their evaluation being of great importance due to the lack of information on this group in Ecuador and the lack of evaluations of high-elevation ecosystems. We highlighted that different species of dung beetles may respond differently to environmental disturbances. Some are considered generalist such as *Uroxys*, which are the most abundant in forest plantations, possibly due to their ability to adapt to disturbed areas. On the other hand, these species may be limited in ecological functions in the ecosystem; due to their limited size, they are inefficient in some functions fulfilled by other large species.

Therefore, these results suggest that anthropogenic change in land use affects the structure of dung beetles, and these findings may help researchers design new studies to evaluate areas with similar altitudes, as well as evaluate whether urban areas affect the results.

## Figures and Tables

**Figure 1 biology-13-00841-f001:**
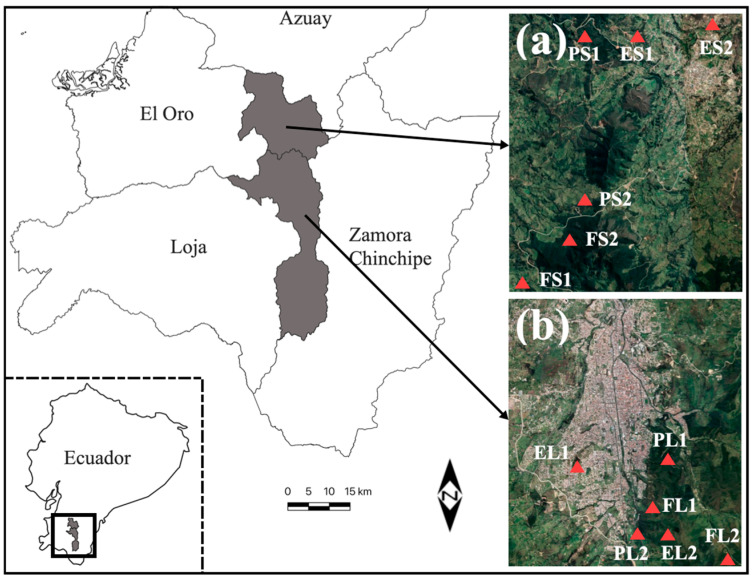
Locations of Loja (**a**). Area 1: Loja Forest (FL), Loja Pinus (PL), Loja Eucalyptus (EL), and Saraguro (**b**). Area 2: Saraguro Forest (FS), Saraguro Pinus (PS), Saraguro Eucalyptus (ES).

**Figure 2 biology-13-00841-f002:**
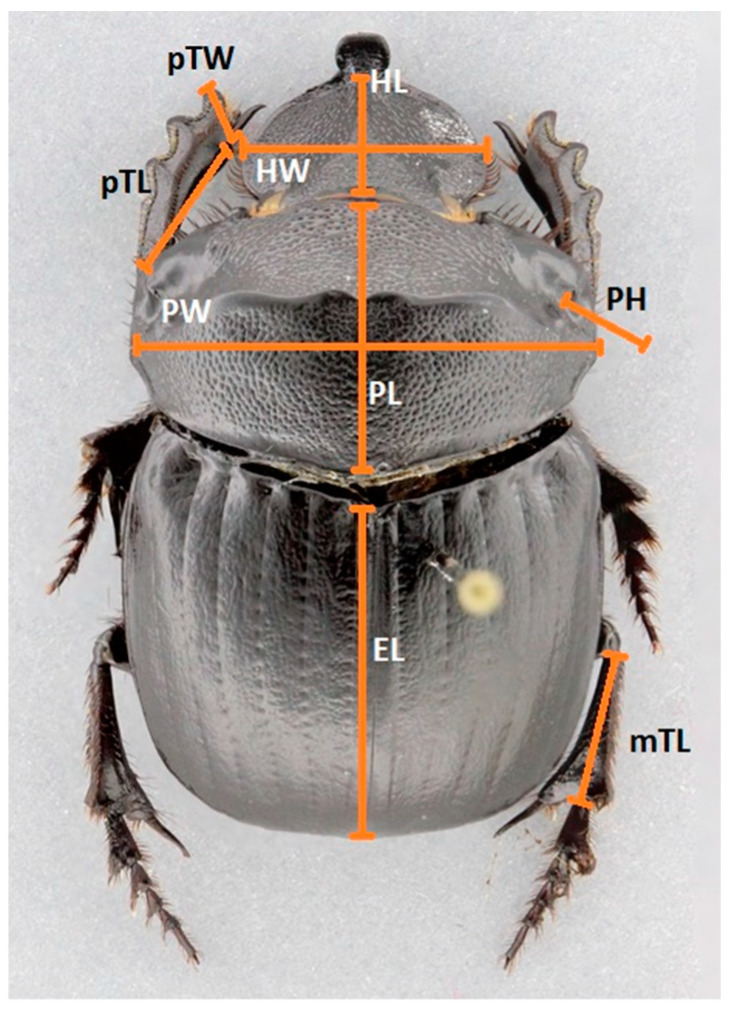
Description of the nine principal traits measured: head width (HW), head length (HL), pronotum width (PW), pronotum length (PL), pronotum height (PH), elytra length (EL), protibia width (pTW), protibia length (pTL), and metatibia length (mTL).

**Figure 3 biology-13-00841-f003:**
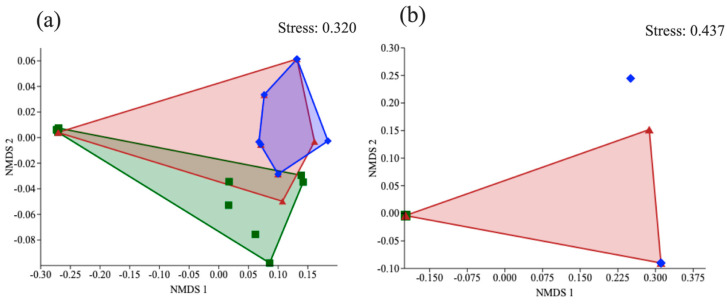
NMDS ordination showed that the dung beetle communities clustered according to land use in two areas: (**a**) Loja and (**b**) Saraguro. Colors: green, forest; red, eucalyptus; blue, pinus.

**Figure 4 biology-13-00841-f004:**
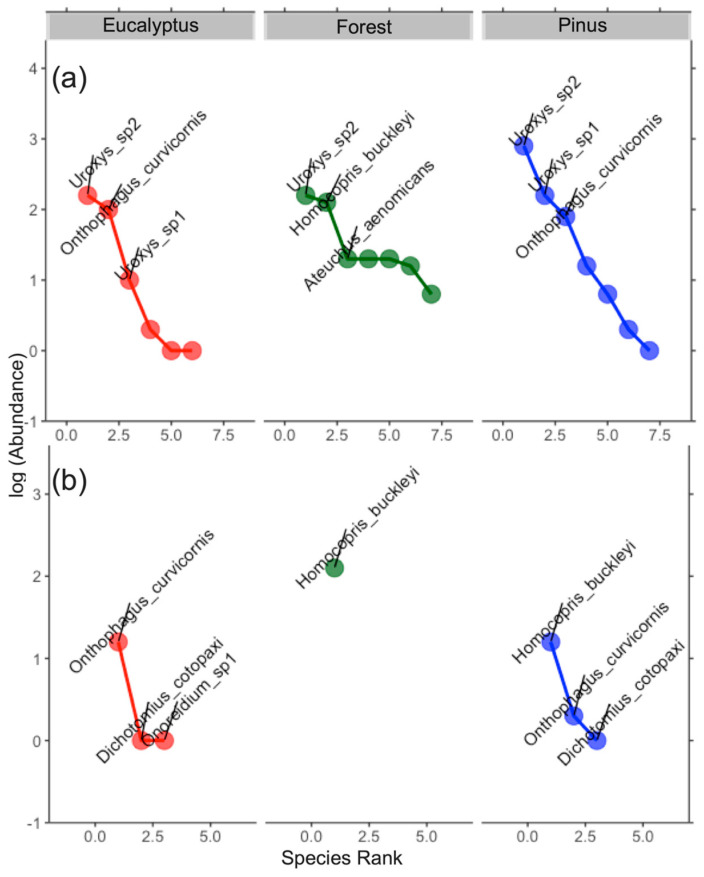
Rank abundance curve of dung beetles in each land use (red = eucalyptus, green = forest and blue = pinus) according to two areas: (**a**) Loja and (**b**) Saraguro.

**Figure 5 biology-13-00841-f005:**
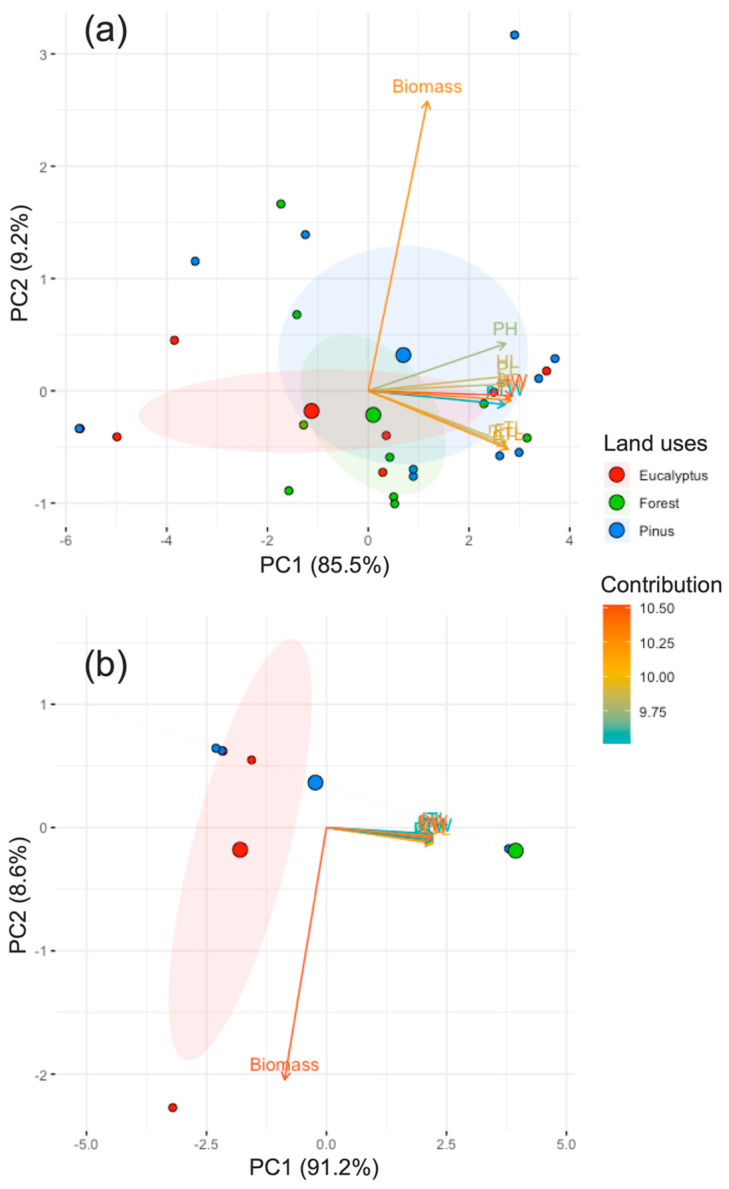
Principal component analysis of 10 morphological traits of dung beetles: head width (HW), head length (HL), pronotum width (PW), pronotum length (PL), pronotum high (PH), elytra length (EL), protibia width (pTW), protibia length (pTL), metatibia length (mTL), biomass, according to areas, (**a**) area 1 (Loja) and (**b**) area 2 (Saraguro), and land uses (eucalyptus, forest and pinus).

**Table 1 biology-13-00841-t001:** Coordinates and altitude per land use, forest, pinus, and eucalyptus, and study area: Loja and Saraguro. For each land use, we selected two sites.

Area	Land Uses	Codes	Coordinates		Altitude (m a.s.l.)
			Latitude	Longitude	
Area 1 (Loja)	Forest	FL1	−4.036	−79.196	2207
		FL2	−4.047	−79.174	2383
	Pinus	PL1	−4.014	−79.192	2220
		PL2	−4.033	−79.196	2218
	Eucalyptus	EL1	−4.015	−79.214	2204
		EL2	−4.034	−79.198	2195
Area 2 (Saraguro)	Forest	FS1	−3.660	−79.268	2963
		FS2	−3.683	−79.270	2944
	Pinus	PS1	−3.615	−79.266	2877
		PS2	−3.652	−79.263	2962
	Eucalyptus	ES1	−3.615	−79.250	2615
		ES2	−3.611	−79.230	2353

**Table 2 biology-13-00841-t002:** Richness and abundance in two study areas in Loja (Area 1): FL (Loja Forest), PL (Loja Pine), EL (Loja Eucalyptus), and Saraguro (Area 2): FS (Saraguro Forest), PS (Saraguro Pine), and ES (Saraguro Eucalyptus).

	Area 1			Area 2		
Species	FL	PL	EL	FS	PS	ES
*Ateuchus aenomicans*	0.7 ± 1.0	-	-	-	-	-
*Cryptocanthon paradoxus*	3.0 ± 5.5	-	-	-	-	-
*Deltochilum robustus*	-	0.3 ± 0.8	-	-	-	-
*Dichotomius cotopaxi*	1.3 ± 1.2	0.8 ± 0.8	0.2 ± 0.4	-	0.06 ± 0.24	0.06 ± 0.24
*Homocopris buckleyi*	-	-	-	6.94 ± 4.01	0.94 ± 1.30	-
*Onoreidium* aff. *cristatum*	-	0.2 ± 0.4	0.2 ± 0.4	-	-	-
*Onoreidium ohausi*	-	-	-	-	-	0.06 ± 0.24
*Onthophagus curvicornis*	3.0 ± 4.7	12.0 ± 7.5	15.0 ± 4.5	-	0.11 ± 0.32	0.83 ± 1.95
*Uroxys* sp. 1	2.8 ± 3.0	25.5 ± 24.0	1.8 ± 1.8	-	-	-
*Uroxys* sp. 2	28.0 ± 34.8	120.5 ± 129.0	24.8 ± 51.3	-	-	-
Richness	6	6	5	1	3	3
Abundance	249	956	250	125	19	17
Richness—area	8			4		
Abundance—area	1455			161		

**Table 3 biology-13-00841-t003:** Coverage-based rarefaction/extrapolation with 95% confidence intervals. Associating dung beetle communities with land uses (forest, pinus and eucalyptus) in two areas (Loja and Saraguro) according to Hill numbers, q0: species richness, q1: Shannon diversity and q2: Simpson diversity in the south of Ecuador.

	Area 1			Area 2		
	FL	PL	EL	FS	PS	ES
Species richness (q0)	7	7.5	7.99	1	3.47	2
CI	7–7	7–10.10	6–12.79	1–1	3–5.80	2–2.93
Shannon diversity (q1)	4	2.29	2.46	1	1.85	1.31
CI	3.65–4.37	2.14–2.45	2.22–2.70	1–1	1.10–2.60	0.84–1.78
Simpson diversity (q2)	3.06	1.72	2.15	1	1.41	1.14
CI	2.77–3.35	1.62–1.83	2–2.29	1–1	0.93–1.90	0.80–1.48
Sample coverage (%)	100	99	99	100	95	100

**Table 4 biology-13-00841-t004:** Effects of land use (forest, eucalyptus, and pinus) on the abundance of dung beetles in two areas (area 1 = Loja, area 2 = Saraguro). Significant values (*p* < 0.05) are highlighted in bold.

Area	Response Variable	Explanatory Variable	Standard Error	Z-Value	*p*-Value
Area 1	Abundance	Land Uses			
		(Intercept)	0.065	55.856	**<0.001**
		Eucalyptus	0.091	0.773	0.439
		Pinus	0.073	19.323	**<0.001**
	Chi test:				
		Land uses			**<0.001**
Area 2	Abundance	Land Uses			
		(Intercept)	0.089	33.950	**<0.001**
		Eucalyptus	0.258	−6.149	**<0.001**
		Pinus	0.246	−6.910	**<0.001**
	Chi test:				
		Land uses			**<0.001**

**Table 5 biology-13-00841-t005:** Effects of land use, traits, and species on morphological traits. Significant values (*p* < 0.05) are indicated in bold.

Area	Variables	Standard Error	*t*-Value	*p*-Value
Area 1	Land uses			
	(Intercept)	−0.062	−5.475	**<0.001**
	Forest	0.004	0.660	0.580
	Pinus	0.015	2.524	0.157
	Traits			
	Elytra length (EL)	0.008	53.663	**<0.001**
	Head length (HL)	0.008	23.058	**<0.001**
	Head width (HW)	0.008	32.493	**<0.001**
	Metatibia length (mTL)	0.008	23.512	**<0.001**
	Pronotum high (PH)	0.008	28.123	**<0.001**
	Pronotum length (PL)	0.008	35.507	**<0.001**
	Protibia length (pTL)	0.008	24.234	**<0.001**
	Protibia width (pTW)	0.008	8.389	**<0.001**
	Pronotum width (PW)	0.008	55.190	**<0.001**
	Species			
	*Cryptocanthon paradoxus*	−0.054	−4.577	**<0.001**
	*Deltochilum robustus*	0.429	21.375	**<0.001**
	*Dichotomius cotopaxi*	0.395	33.996	**<0.001**
	*Onoreidium cristatum*	−0.067	−3.336	**<0.001**
	*Onthophagus curvicornis*	0.069	7.308	**<0.001**
	*Uroxys* sp. 1	0.060	6.415	**<0.001**
	*Uroxys* sp. 2	−0.003	−0.307	0.759
Area 2	Land uses			
	(Intercept)	0.0670	3.931	**<0.001**
	Forest	0.0580	0.628	0.530
	Pinus	0.0501	0.562	0.575
	Traits			
	Elytra length (EL)	0.048	13.589	**<0.001**
	Head length (HL)	0.048	5.651	**<0.001**
	Head width (HW)	0.048	9.056	**<0.001**
	Metatibia length (mTL)	0.048	5.106	**<0.001**
	Pronotum high (PH)	0.048	8.687	**<0.001**
	Pronotum length (PL)	0.048	9.488	**<0.001**
	Protibia length (pTL)	0.048	4.544	**<0.001**
	Protibia width (pTW)	0.048	0.686	0.493
	Pronotum width (PW)	0.048	15.682	**<0.001**
	Species			
	*Homocopris buckleyi*	0.064	−1.515	0.130
	*Onoreidium ohausi*	0.096	−0.118	0.907
	*Onthophagus curvicornis*	0.060	−6.507	**<0.001**

## Data Availability

The data presented in this study are available on request from the corresponding author.

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
