# Peer review of "The Impact of Eucalyptus and Pine Plantations on the Taxonomic and Functional Diversity of Dung Beetles (Coleoptera: Scarabaeidae) in the Southern Region of Ecuador"

_biology, 2024, doi:10.3390/biology13100841_

Round 1
Reviewer 1 Report
Comments and Suggestions for Authors
Dear Authors,
the present paper is devoted to study of affected areas recover and impact of human activity to the scarab beetles communities in newly planted or organised forests in Ecuador. All facilities aimed to protection of nature or recover disturbed territories are important and actual nowadays. The study is well organised and results are correct and well-reasoned, the manuscript is recommended for publication. Some questions appeared after acquaintance with the manuscript. Will be nice if the authors will find possible to make explanations in the text after familiarisation with the questions.
1. Why pitfall traps were chosen for scarab beetles accounting? This type of traps are usually used to reveal herpetobiont beetles, and a lot of other trap types are known strictly for dung beetles catching with high effectiveness?
2. Specifics of species compositions occur in studied types of forest is not discussed, and we can not imagine what species are predominate in native forest and what species are typical for each type of the forest studied.
3. When analyzed diversity in eucalypti and pine tree forest you did not mention high concentration of volatile oils produced by trees and prevent generation of microbiota, which providing successful decaying of excrements and kept condition of substrate availability for beetles. Why this factor did not consider in the work?
4. Some terms in conclusions look self-evident, namely: “Therefore, these results suggest that anthropogenic change in land use affects the structure of dung beetles, however, these findings may help researchers to design new studies to evaluate in areas with similar altitudes, as well as evaluate whether the urban area affects the results”. Honestly, discussion of concrete impact of newly planted or organized forests to beetle communities was expectable within this study, unfortunately, we are readdressed to future investigation instead.
Thank you for interesting study and good luck with your further work
Author Response
The present paper is devoted to study of affected areas recover and impact of human activity to the scarab beetles communities in newly planted or organised forests in Ecuador. All facilities aimed to protection of nature or recover disturbed territories are important and actual nowadays. The study is well organised and results are correct and well-reasoned, the manuscript is recommended for publication. Some questions appeared after acquaintance with the manuscript. Will be nice if the authors will find possible to make explanations in the text after familiarisation with the questions.
- Why pitfall traps were chosen for scarab beetles accounting? This type of traps are usually used to reveal herpetobiont beetles, and a lot of other trap types are known strictly for dung beetles catching with high effectiveness?
Response 1: Thanks for pointing this out, pitfall traps were selected based on the meta-analysis researched by Mora, which indicates that pitfall traps are the dominant method for capturing dung beetles. Line 126-128.
- Specifics of species compositions occur in studied types of forest is not discussed, and we can not imagine what species are predominate in native forest and what species are typical for each type of the forest studied.
Response 2: According to your comment, therefore, the IndVal index was used in our research for the analysis of indicator species. Line 169-171.
- When analyzed diversity in eucalypti and pine tree forest you did not mention high concentration of volatile oils produced by trees and prevent generation of microbiota, which providing successful decaying of excrements and kept condition of substrate availability for beetles. Why this factor did not consider in the work?
Response 3: The point you mention is very relevant, our research is focused on the analysis of the taxonomic and functional diversity of dung beetles, however, it would be interesting to study the chemical interaction between forest plantations and soil. Line 70-71.
- Some terms in conclusions look self-evident, namely: “Therefore, these results suggest that anthropogenic change in land use affects the structure of dung beetles, however, these findings may help researchers to design new studies to evaluate in areas with similar altitudes, as well as evaluate whether the urban area affects the results”. Honestly, discussion of concrete impact of newly planted or organized forests to beetle communities was expectable within this study, unfortunately, we are readdressed to future investigation instead.
Thank you for interesting study and good luck with your further work
Response 4: We agree with your observation, we started with the same hypothesis mentioned, however, it is important to note that for Ecuador we do not have published data focused on our topic, that is why this research is relevant and can serve as a starting point for future research that want to delve into this topic. Line 424-428.
Reviewer 2 Report
Comments and Suggestions for Authors
The data of the manuscript are analyzed systematically and comprehensively. The manuscript is clearly structured. I have only some small suggestions for the manuscript.
In Line 46, [11] suggest that, you should add the author.
In Line 51, according to [15], you should add the author.
In Materials and Methods, in Line 132 and 133, 7. pTL= length of the protibia; 8. pTW= width of the protibia, and 9. mTL= length of metatibia, protibia may be fore tibia, metatibia may be hind tibia, according to Linz DM, Hu Y, Moczek AP. 2019 The origins of novelty from within the confines of homology: the developmental evolution of the digging tibia of dung beetles. Proc. R. Soc. B 286: 20182427.
In Seibold et al. (2019), they use body length of arthropod. In Line 153, you didn’t measure the body length indirectly, and you think head length + pronotum length + elytra length = body length. Whether this method will bring systematic error?
In line 191, 6,94 ± 4,01, 6.94 ± 4.01? Check the data.
In line 219, p>0.001 or p<0.001? In line 220, p> 0.002 or p< 0.002? In line 238 and 239, also check the p values.
Author Response
In Line 46, [11] suggest that, you should add the author.
In Line 51, according to [15], you should add the author.
Response 1: Thank you for pointing this out, the correction was made. Line 46 and 51.
In Materials and Methods, in Line 132 and 133, 7. pTL= length of the protibia; 8. pTW= width of the protibia, and 9. mTL= length of metatibia, protibia may be fore tibia, metatibia may be hind tibia, according to Linz DM, Hu Y, Moczek AP. 2019 The origins of novelty from within the confines of homology: the developmental evolution of the digging tibia of dung beetles. Proc. R. Soc. B 286: 20182427.
Response 2: Thank you for pointing this out, however, we are using the standardized names.
In Seibold et al. (2019), they use body length of arthropod. In Line 153, you didn’t measure the body length indirectly, and you think head length + pronotum length + elytra length = body length. Whether this method will bring systematic error?
We agree with your comment, there could be a systematic error due to morphological variability, differences in posture and body flexibility, so in our research we estimated an average error to take this error into account.
In line 191, 6,94 ± 4,01, 6.94 ± 4.01? Check the data.
Response 4: Thank you for pointing this out, the correction was made. Line 198.
In line 219, p>0.001 or p<0.001? In line 220, p> 0.002 or p< 0.002? In line 238 and 239, also check the p values.
Response 5: Thank you for pointing this out, the correction was made. Lines 226-227.
Reviewer 3 Report
Comments and Suggestions for Authors
This is a relatively good manuscript, but the Title is misleading: I can not see the impact of these plantations in your manuscript. Based on the content, it likely pertains to dung beetle taxonomic and functional diversity in Ecuador under different land-use scenarios. Please consider a more to-the-point title, that has to do with the taxonomic and functional diversity of dung beetles in Southern Ecuador, and a possible subtitle regarding the effects of land use in high elevation ecosystems.
The introduction provides necessary context on the importance of dung beetles in ecosystem functioning, particularly in nutrient cycling, and the need to assess the effects of land-use change on this group. However, it could benefit from a more robust literature review. Please, include more specific examples from other studies in similar ecosystems. Also, the study’s goals seem to be implied but could be more explicitly stated.
The methodology appears thorough, though only parts were shared. It should clearly outline the criteria for selecting study sites (why Loja and Saraguro?), the sampling design, including details on the duration and frequency of beetle collection, and the statistical analyses applied to assess diversity, functional traits, and NMDS results.
The results section is detailed and informative. However, you should explain why, in your opinion, dung beetle communities overlapped in plantations in both areas but were distinct in forests.
In the discussion, consider adding a paragraph addressing the implications of the findings for conservation. Since some species were specialists, such as Homocopris buckleyi, how should forest managers account for this in conservation planning? There are plenty of silviculture research papers that you can refer. Also, a more detailed comparisons to other studies in similar ecosystems (that I told you to add in the introduction) would enrich the discussion.
In general, the manuscript offers valuable insights into the taxonomic and functional diversity of dung beetles in a highly biodiverse and ecologically important region of Ecuador. The study is timely, addressing how land-use changes, particularly the introduction of forest plantations, affect biodiversity and ecosystem services. This is relevant for both conservation biology and land management practices.
As a last minor issue, the way you cite [11] (Line 46), [15] (Line 51, [18] (Lines 56 and 62) and [29-30] (Line 124), does not follow the manuscript's standards. You will have to use the name of the writer, and then use the citation.
Comments on the Quality of English Language
Minor grammatical and typographical corrections throughout the manuscript are needed. For example, Line 196 Uroxys sp. 2 represented the 65 % of total abundance 197 and present only in Area 1 (Loja) needs to be fixed. Also, in lines 206 and 242, please change pinus plantation to pine plantation (or if you prefer the genus name, write it correctly : Pinus, but also cahnge Eucalyptus as well).
Author Response
This is a relatively good manuscript, but the Title is misleading: I can not see the impact of these plantations in your manuscript. Based on the content, it likely pertains to dung beetle taxonomic and functional diversity in Ecuador under different land-use scenarios. Please consider a more to-the-point title, that has to do with the taxonomic and functional diversity of dung beetles in Southern Ecuador, and a possible subtitle regarding the effects of land use in high elevation ecosystems.
Thank you for your thoughtful feedback and for taking the time to review our manuscript. We appreciate your suggestion regarding the title. However, after careful consideration, we believe that the current title aligns with the broader scope of the study, which encompasses both the taxonomic and functional diversity of dung beetles and the ecological impacts of different plantation types.
The plantations and land-use changes are integral to our analysis, and the title reflects our aim to assess how these practices influence biodiversity and ecosystem functions. We feel that this focus is essential to understanding the broader environmental implications.
We hope this explanation clarifies our approach, and we look forward to any further feedback you may have.
The introduction provides necessary context on the importance of dung beetles in ecosystem functioning, particularly in nutrient cycling, and the need to assess the effects of land-use change on this group. However, it could benefit from a more robust literature review. Please, include more specific examples from other studies in similar ecosystems. Also, the study’s goals seem to be implied but could be more explicitly stated.
Response 1: We agree with your comment, therefore, we are looking for other examples to complete the information. Line: 51-52.
The methodology appears thorough, though only parts were shared. It should clearly outline the criteria for selecting study sites (why Loja and Saraguro?), the sampling design, including details on the duration and frequency of beetle collection, and the statistical analyses applied to assess diversity, functional traits, and NMDS results.
According with this comment. We have taken into account the criteria for selecting both sites (Loja and Saraguro).
Line 77-79: The research was carried out in two of the areas with the greatest distribution of pine and eucalyptus plantations in southern Ecuador.
Line 81-83: We include “These species were used for this study because they are the two main species that have been reforested since the 1970s.”
Line 122: the frequency of sampling was mentioned, it is 48 hours per site.
The results section is detailed and informative. However, you should explain why, in your opinion, dung beetle communities overlapped in plantations in both areas but were distinct in forests.
Line 374-388: we include Several factors may account for the similarity of dung beetle communities in plantations across both regions, in contrast to the unique communities observed in forests. The homogeneous nature of eucalyptus and pine monocultures in plantations often results in reduced habitat diversity [47], creating comparable ecological conditions across different plantation sites. This uniformity may support similar dung beetle species in both areas. Plantations typically lack the complex structure, microhabitats, and diverse plant associations characteristic of forests, leading to a convergence of species adapted to simplified environments [48,49]. Forests, on the other hand, offer a broader range of niches due to their greater structural diversity, including variations in plant species composition, canopy coverage, understory complexity, and availability of resources like decomposing organic matter [50]. These factors likely create distinct microenvironments that support more specialized dung beetle communities in each forest area. As a result, forest dung beetle assemblages remain more distinct due to the variability in habitat conditions and resource availability, while the more uniform conditions in plantations lead to community overlap.
In the discussion, consider adding a paragraph addressing the implications of the findings for conservation. Since some species were specialists, such as Homocopris buckleyi, how should forest managers account for this in conservation planning? There are plenty of silviculture research papers that you can refer. Also, a more detailed comparisons to other studies in similar ecosystems (that I told you to add in the introduction) would enrich the discussion.
In general, the manuscript offers valuable insights into the taxonomic and functional diversity of dung beetles in a highly biodiverse and ecologically important region of Ecuador. The study is timely, addressing how land-use changes, particularly the introduction of forest plantations, affect biodiversity and ecosystem services. This is relevant for both conservation biology and land management practices.
Lines 375-408: We include Several factors may account for the similarity of dung beetle communities in plantations across both regions, in contrast to the unique communities observed in forests. The homogeneous nature of eucalyptus and pine monocultures in plantations often results in reduced habitat diversity [49], creating comparable ecological conditions across different plantation sites. This uniformity may support similar dung beetle species in both areas. Plantations typically lack the complex structure, microhabitats, and diverse plant associations characteristic of forests, leading to a convergence of species adapted to simplified environments [50,51]. Forests, on the other hand, offer a broader range of niches due to their greater structural diversity, including variations in plant species composition, canopy coverage, understory complexity, and availability of resources like decomposing organic matter [52]. These factors likely create distinct microenvironments that support more specialized dung beetle communities in each forest area. As a result, forest dung beetle assemblages remain more distinct due to the variability in habitat conditions and resource availability, while the more uniform conditions in plantations lead to community overlap.
Species with specific requirements, such as Homocopris buckleyi, rely heavily on particular environmental conditions or resources. The occurrence of these organisms in wooded regions suggests that these environments offer unique ecological niches supporting species with limited habitat flexibility [45]. As a result, protecting these specialists is vital, given that their existence hinges on the conservation of specific ecosystems. Research indicates that forests, especially those at higher elevations, contain distinct dung beetle communities, including specialist species [53,54]. These discoveries underscore the importance of safeguarding forest habitats, particularly those that sustain specialist organisms. Moreover, it is crucial to maintain connections between forest fragments to facilitate movement and genetic exchange among specialist species populations, preventing isolation and localized extinctions [55]. Conservation strategies should be implemented to preserve habitat diversity, ensuring the protection of microhabitats and preventing human activities like logging or land conversion from reducing ecological complexity [56–58]. Promoting the use of agroforestry practices or more diverse plantation management techniques that incorporate native species can enhance biodiversity and lessen the negative effects of monoculture plantations on specialist species [59,60]. Establishing ongoing biodiversity monitoring programs that focus on dung beetle communities, especially specialist species, is essential to ensure the effectiveness and sustainability of conservation efforts [31,61].
As a last minor issue, the way you cite [11] (Line 46), [15] (Line 51, [18] (Lines 56 and 62) and [29-30] (Line 124), does not follow the manuscript's standards. You will have to use the name of the writer, and then use the citation.
This has already been corrected
Comments on the Quality of English Language
Minor grammatical and typographical corrections throughout the manuscript are needed. For example, Line 196 Uroxys sp. 2 represented the 65 % of total abundance 197 and present only in Area 1 (Loja) needs to be fixed. Also, in lines 206 and 242, please change pinus plantation to pine plantation (or if you prefer the genus name, write it correctly : Pinus, but also change Eucalyptus as well).
Line 204-205: we include In Area 1 (Loja), Uroxys sp. 2 constituted 65% of the total abundance and was exclusively found in this location